



# Open-source, 3D-printed, high-pressure (50 bar) liquid-nitrogen-cooled *para*-hydrogen generator

Frowin Ellermann[1], Andrey Pravdivtsev[1], Jan-Bernd Hövener[1]

[1] Section for Biomedical Imaging, Molecular Imaging North Competence Center (MOIN CC), Department of Radiology and
Neuroradiology, University Medical Center Schleswig-Holstein (UKSH), Kiel, 24118, Germany

*Correspondence to*: Frowin Ellermann (frowin.ellermann@rad.uni-kiel.de), Jan-Bernd Hövener (jan.hoevener@rad.uni-kiel.de)

**Abstract.** The signal of magnetic resonance imaging (MRI) can be enhanced by several orders of magnitude using
hyperpolarization. In comparison to a broadly used Dynamic Nuclear Polarization (DNP) technique that is already used in clinical trials, the *para*-hydrogen ($p$H$_2$) based hyperpolarization approaches are less cost-intensive, scalable and offer high throughput. However, a $p$H$_2$ generator is necessary. Available commercial $p$H$_2$ generators are relatively expensive (10,000 – 150,000 €). To facilitate the spread of $p$H$_2$ hyperpolarization studies, here, we provide the blueprints and 3D-models as open-source for a low-cost (<3,000 €) 50 bar liquid nitrogen $p$H$_2$ generator.

## 15 1 Introduction

Nuclear Magnetic Resonance (NMR), as well as Magnetic Resonance Imaging (MRI), are widely used in medical imaging and chemical analysis. Despite the great success of these techniques (Feyter et al., 2018; Lange et al., 2008; Watson et al., 2020), the low signal-to-noise ratio of NMR limits promising applications such as in vivo spectroscopy or imaging nuclei other than $^1$H (Wilferth et al., 2020; Xu et al., 2008). The hyperpolarization of nuclear spins boosts the signal of selected molecules
by orders of magnitude. This way, imaging of the lung or metabolism has become feasible (Beek et al., 2004; Kurhanewicz et al., 2011).

Among techniques, Parahydrogen And Synthesis Allows Dramatically Enhanced Nuclear Alignment (PASADENA) (Bowers and Weitekamp, 1986, 1987; Eisenschmid et al., 1987) has found application from catalysis research to metabolic imaging (Hövener et al., 2018; Kovtunov et al., 2018).

The production of *para*-hydrogen ($p$H$_2$) is relatively easy: H$_2$ gas is flowing through a catalyst at cold temperatures; maximum *para*-enrichment of almost 100 % is achieved at about 25 K (Gamliel et al., 2010; Jeong et al., 2018; Kiryutin et al., 2017). To reach low temperatures, hence enrich $p$H$_2$, liquid cryogens (Buckenmaier et al., 2018; Jeong et al., 2018) or electric cryopumps (Feng et al., 2012) are used. Electronic setups were reported, e.g. for pressures up to 50 bar of ≈ 100 % $p$H$_2$ (Hövener et al.,





2013). Liquid nitrogen ($lN_2$)-based systems were described, however, often with limited description, low production rate and
pressure.

Thus, in this contribution, we report a *para*-hydrogen generator (PHG) based on $lN_2$ that operates at a pressure of up to 50 bar at a cost of less than 3000 €. The setup is easy to replicate as it is fully open-sourced (Ellermann, 2020b) and all parts are either off-the-shelf, 3D-printed or can be constructed easily. Besides, we introduce an automated $pH_2$ quantification method using a 1 T benchtop NMR and Arduino-based process control.

**Background** In 1933 Werner Heisenberg received his Nobel Prize "for the creation of quantum mechanics, the application of which has, inter alia, led to the discovery of the allotropic forms of hydrogen" (NobelPrize.org, 2020). Allotropy is a property of substances to exist in several forms, in the same physical state. Two forms of hydrogen usually are referred to as spin isomers; they are *para*-hydrogen ($pH_2$) and *ortho*-hydrogen ($oH_2$). Hydrogen is not the only compound that has stable or long-lived spin isomers at room temperature (rt) there are many examples: deuterium (Knopp et al., 2003), water (Mammoli et al.,
2015; Meier et al., 2015), ethylene (Zhivonitko et al., 2013), and even naphthalene derivative (Stevanato et al., 2015).

The spin of hydrogen nuclei (proton) is the origin of the two allotropic forms or two spin isomers of hydrogen. Protons have spin-½, hence they are fermions. Fermions are particles that follow Fermi-Dirac statistics, therefore the sign of the total wave function of $H_2$ has to change when two nuclei are exchanged. The spin space of two spin-½ consists of $\left(2 \cdot \frac{1}{2} + 1\right)^2 = 4$ states. They are three symmetric spin states: $|T_+\rangle = |\alpha\alpha\rangle$, $|T_0\rangle = (|\alpha\beta\rangle + |\beta\alpha\rangle)/\sqrt{2}$, $|T_-\rangle = |\beta\beta\rangle$ and one asymmetric nuclear spin
state $|S\rangle = (|\alpha\beta\rangle - |\beta\alpha\rangle)/\sqrt{2}$ (Fig. 1). Here conventionally $|\alpha\rangle$ and $|\beta\rangle$ states are nuclei spin states with the projection of spin on 0Z axis ½ and -½, $|T_+\rangle$, $|T_0\rangle$ and $|T_-\rangle$ are triplet spin states of two spin-½ with a total spin 1 and the projection on 0Z axis +1, 0 and -1, and $|S\rangle$ is a singlet spin state of two spin ½ with the total spin 0.

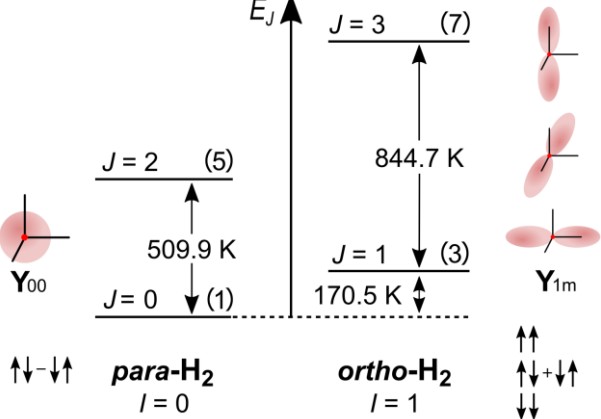

**Figure 1: The rotational energy level diagram for isolated $H_2$.** The angular distribution of the two lowest rotational states ($Y_{00}$ corresponds
to $J = 0$ and $Y_{1+1} \pm Y_{1-1}$ and $Y_{10}$ corresponds to $J = 1$) and spin states of *ortho-* and *para*-hydrogen are indicated. The numbers in parenthesizes are the degeneracies of the state $2J + 1$. The energy of rotation spin states in units of K is equal to $E_J = J(J + 1)\theta_R$ with $\theta_R =$





87.6 K (Atkins and De Paula, 2006). The distance between two adjacent energy levels is $E_{J+1} - E_J = 2(J+1)\theta_R$. The figure was inspired by an illustration of I. F. Silvera (1980).

The rotational wave function after nuclei permutation does not change, because of the molecular symmetry, and is only multiplied by $(-1)^J$, with $J$ being the rotational quantum number of the state. Hence, $H_2$ with a symmetric nuclear spin state (triplet states) can have only an asymmetric rotational state ($J$ is odd); such $H_2$ is called $oH_2$. And vice versa, $H_2$ with an asymmetric nuclear spin state (singlet state) can have only symmetric rotation states ($J$ is even); such $H_2$ is called $pH_2$.

The difference in the energy levels of two ground states of *ortho* ($J = 1$) and *para* ($J = 0$) hydrogen is $E_{J=1} - E_{J=0} = 2\theta_R \cong$

175 K (Fig. 1) (Atkins and De Paula, 2006). Such a big energy gap allows a relatively simple way of spin-isomer enrichment: for $H_2$ the ground state is $pH_2$ and its population can be increased by cooling down the gas (Fig. 2) (M. Richardson et al., 2018). The ratio of the number of molecules of $pH_2$, $n_{pH_2}$, to $oH_2$, $n_{oH_2}$, in thermal equilibrium is given by Boltzmann distribution of rotational energy levels:

$$\frac{n_{pH_2}}{n_{oH_2}} = \frac{\sum_{J=even\geq0}(2J+1)\exp(-J(J+1)\theta_R/T)}{3\sum_{J=odd\geq1}(2J+1)\exp(-J(J+1)\theta_R/T)}. \tag{1}$$

Since only two allotropic states of $H_2$ exist, their fractions can be easily obtained: $f_{pH_2} = \frac{n_{pH_2}}{n_{pH_2}+n_{oH_2}} = \frac{1}{1+\frac{n_{oH_2}}{n_{pH_2}}}$ and $f_{oH_2} = \frac{1}{1+\frac{n_{pH_2}}{n_{oH_2}}}$. At room temperatures ($T \cong 298$ K) $n_{pH_2} : n_{oH_2}$ is close to $3:1$, at 77 K – the normal boiling point of nitrogen – the ratio is close to $1:1$, and at 25 K $f_{pH_2} \cong 98\%$ (Fig. 2).

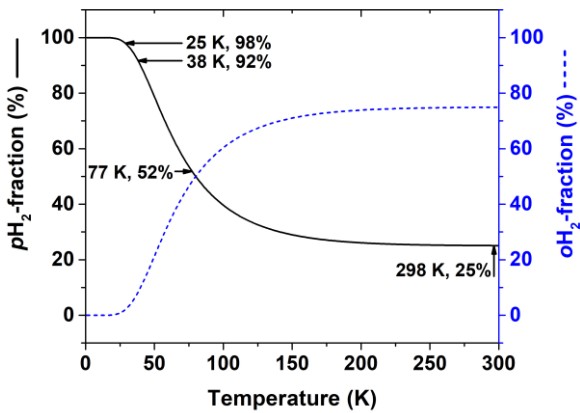

**Figure 2: Thermal equilibrium fractions of $pH_2$ and $oH_2$ as a function of temperature calculated with Eq. 1 and $J = 6$.** Four
temperatures are marked: (1) 298 K – "room temperature", $f_{pH_2} \cong 25$ %, (2) 77 K – the boiling temperature of liquid nitrogen, $f_{pH_2} \cong$ 52 %, (3) 38 K – medium conversion temperature of Bruker $pH_2$ generator, $f_{pH_2} \cong 92$ %, (4) 25 K – conversion temperature of high-pressure PHG, $f_{pH_2} \cong 98 \pm 2$ % (Hövener et al., 2013).





**Technology review**

$p$H$_2$ fraction, $f_{pH_2}$, of 90 % and above is produced by PHGs with single- or dual-stage cryostats run by Helium compressors. A single-stage cryostat was reported to operate at 36-40 K with a flow rate of 0.2 SLM (Standard Liter per Minute), 10 bar maximum delivery pressure, and $f_{pH_2} \cong 85 - 92$ % (Bruker, Billerica, U.S.A.); dual-stage cryostats operate at temperatures below 25 K where $f_{pH_2}$ reaches 100 % (note that the boiling point of H$_2$ is 21 K) (Haynes, 2011). All these PHGs were specifically designed with PHIP (*Para*-Hydrogen Induced Polarization) in mind; meaning for a relatively low scale of

production and in-lab use (not for the industry). These setups required some on-site assembling and were realized in different designs, e.g. with pulsed injection (Feng et al., 2012) or continuous flow (Hövener et al., 2013). The continuous flow setup was reported to operate at a conversion temperature of 25 K, 4 SLM flow rate, 50 bar maximum delivery pressure, and experimentally obtained $f_{pH_2} \cong 98 \pm 2$ % (Hövener et al., 2013).

These setups work reliably and don't require much in terms of service (e.g. no liquid cryogens). Disadvantages, however,

include high initial investments (40.000 – 150.000 €), some maintenance of the He-compressor and cryostat (≈ 10.000 € every 25.000 h operational time), some site requirements (~4 kW cooling water, appropriate safety precautions) and operational cost in form of electricity (>4 kW electrical power) (Table 1).

A 100 % $p$H$_2$ enrichment, however, may not always be needed. 50 % $p$H$_2$ fraction provides already 1/3 of the maximum polarization at 1/10 of the costs (or less) (M. Richardson et al., 2018). To achieve $f_{pH_2}$ of 50 % the temperature of lN$_2$, 77 K,

is sufficient. Indeed, much of the initial studies were performed with lN$_2$-based PHGs - and still are (Kiryutin et al., 2017; Meier et al., 2019). The design of such PHGs is generally simple - a catalyst chamber or tube immersed in lN$_2$. But just like cryostat-based PHGs, lN$_2$-based PHGs are continuously improving. As such, recent advances included remarkable work, where 20 l lN$_2$ were sufficient to provide $p$H$_2$ continuously for 20 days (Jeong et al., 2018).

Interestingly, in various cases it was demonstrated that increased flow rate and pressure of $p$H$_2$ can boost the signal of PHIP

or Signal Amplification By Reversible Exchange (SABRE)(Adams et al., 2009; Rayner and Duckett, 2018) way beyond the factor of 3 offered by PHGs with close to $f_{pH_2} \cong 100\%$ (Colell et al., 2017; Rayner et al., 2017; Štěpánek et al., 2019; Truong et al., 2015).





**Table 1: Performance comparison of several PHGs:** (1) Bruker PHG 90, (2) dual-stage cryostats (DSC) (Hövener et al., 2013), (3) a pulsed PHG (Feng et al., 2012) (4) U-shape PHG (Kiryutin et al., 2017), (5) HyperSpin-PHG (Meier et al., 2019), (6) economical PHG (Jeong et al., 2018), (7) glass-trap PHG (Gamliel et al., 2010) and (8) in house designed and build PHG (this work). Given prices include all connectors, cylinders, and 19 % VAT. $lN_2$ stands for liquid nitrogen and "cc-He" for closed-cycle He compressor.

| | Name | Operating temperature (K) [method] | $f_{pH_2}(\%)$ | Initial flow rate (SLM) | Max. pressure (bar) | Price (€) |
|---|---|---|---|---|---|---|
| 1 | Bruker BPHG 90 | 36-40 [cc-He] | 85-92 | ≤ 0.2 | 10 | 100,000–150,000 |
| 2 | DSC (Hövener et al., 2013) | 25 [cc-He] | 98 ± 2 | 4 | 50 | 37,000 |
| 3 | Pulsed PHG (Feng et al., 2012) | 15 [cc-He] | 98 | 0.9 | 20 | N.A. |
| 4 | HyperSpin-PHG (Meier et al., 2019) | 20-77 [cc-He] | N.A.[a] | N.A. | Min. 10 | N.A. |
| 5 | U-shape PHG (Kiryutin et al., 2017) | 77 [$lN_2$] | ~50 | 0.36[b] | Min. 3 | N.A. |
| 6 | Economical PHG (Jeong et al., 2018) | 77 [$lN_2$] | ~50 | N.A. | N.A. | N.A. |
| 7 | Glass-trap PHG (Gamliel et al., 2010) | 77 [$lN_2$] | 46.3 ± 1.3 | 0.0025[c] | ~1 | N.A. |
| 8 | This work | 77 [$lN_2$] | 51.6 ± 0.88 | 2.0[d] | 50[e] | 2,988[f] |

## 2 Methods

### 2.1 3D design of PHG

The principal scheme of $lN_2$ base of a complete PHG consists of $H_2$ gas supply, generator and $pH_2$ storage (Fig. 3). A model of the PHG was designed (Autodesk Inventor 2019, San Rafael, U.S.A.). Aluminium profiles and steel angles (30 mm, Bosch Rexroth, Stuttgart, Germany) were used to construct the chassis. Copper tubes (OD 6 mm, ID 4 mm, rated for 229 bar, R220, Landefeld, Kassel-Industriepark, Germany) and valves (Swagelok, Solon, U.S.A.) were mounted on the chassis using 3D-printed parts (Ultimaker PLA "Perlweiss" Filament, Ultimaker S5, Ultimaker Cura, Utrecht, Netherlands). A 2 L stainless steel dewar was placed in the chassis (DSS 2000, 2 L, KGW Isotherm, Karlsruhe, Germany). The same copper tubes were used to wind a coil with 5.4 turns and a diameter of 86 mm. About 1.5 ml granular Iron (III) oxide (371254-50G, Sigma-

---

[a] Depends on choice of coolant

[b] Estimated average flow (3.5 L volume filled to 3 bar in 90 min) calculated by us

[c] Estimated average flow (0.6 L volume filled to 1 bar in 240 min) calculated by us

[d] Highest average flow on filling 1 L bottle to 10 bar without sacrificing enrichment

[e] 50 bar $pH_2$ delivery was tested. Used parts allow pressure of at least 100 bar (safety margin)

[f] including $H_2$-gas sensor and excluding $H_2$ and $N_2$ bottles/regulators





Aldrich, St. Louis, U.S.A.) was filled into the coil. In both ends of the coil, the cotton wool was pressed to keep the catalyst in place and protect the rest of the system from contaminations. All fittings, T-pieces, ball-valves, overpressure-valve, flow

regulators, the pressure gauge, and fast connectors (Swagelok, Solon, U.S.A.) were connected with the same copper tube. For storage of $p$H$_2$, a 1 L cylinder made from aluminium was used (C1, A6341Q, Luxfer, Nottingham, UK). All parts were chosen to be rated for 100 bar or more to allow for a 100 % safety margin. A list of all parts is given in Appendix A. The models of PHG, 3D-printing parts and experimental macros (experimental protocols) are available (Ellermann, 2020b).

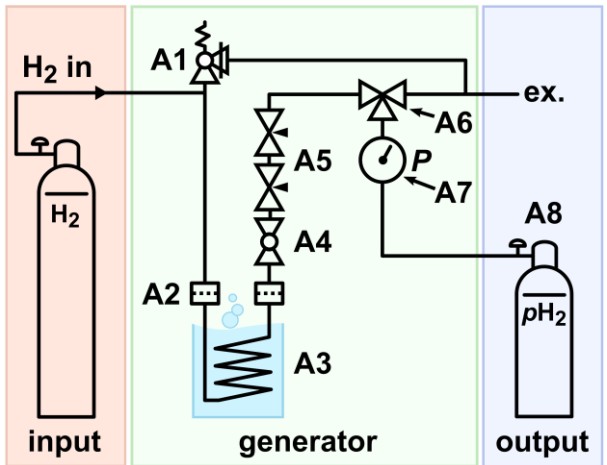


**Figure 3: Schematic view of the PHG.** H$_2$ gas is supplied via the inlet e.g. from a 50 L 200 bar cylinder. The gas flows through a filter (A2) into the *ortho-para*-conversion unit (A3) immersed in lN$_2$, where it is cooling down and thus getting enriched with $p$H$_2$ component. The *para*-hydrogen-enriched gas exits *ortho-para*-conversion unit, warms up and passes another particle filter. The filters reduce the contamination of the setup with catalyst. A ball-valve (A4) is used to start or stop gas flow. Two needle valves (A5) are used to control the

flow rate. A 3-way-valve (A6) allows to fill or drain the storage cylinder. A 100 bar safety valve (A1) is connected to the system to relief potential excess of pressure.

**2.2 Safety concept**

A crucial part of a PHG is the development of a safety concept which includes a detailed risk assessment and comprehensive operating manual. The handling of pressurized H$_2$ gas entails the risk of pressured gases, forming a potentially explosive

mixture with air as well as hydrogen embrittlement (Beeson and Woods, 2003; National Aeronautics and Space Administration, 1997). To reduce these risks, the following safety requirements were set:

1.    Safety by design
      a.    Pressure ratings of parts
            i.    All components, in contact with pressurized gas, are rated for minimum 100 bar

135                ii.    Mechanical pressure gauge
            iii.    100 bar safety valve for overpressure control





        b.   Avoidance of formation of explosive $H_2$-air-mixture and potential ignition

               i.   Reduction of $H_2$ in the system by minimizing the inner volume of the gas lines

               ii.   No electrical components in the system

140                iii.   Avoidance of temperatures above flame point

               iv.   Avoidance of inductive and static spark charges

               v.   High $H_2$ throughput and storage of $pH_2$ in small cylinder leads to a short operating time of PHG

        c.   Easy maintenance due to the simple and open design concept

    2.   Safety by site and operation

145         a.   Strong ventilation in the installation site

        b.   No public access

        c.   Appropriate warning signs

        d.   Usage by trained personnel according to manual only

        e.   Use of safety goggles and safety gloves for the handling of $lN_2$

150         f.   $H_2$ sensor (for leakage alarm at 50 ppm $H_2$ level)

        g.   Regular inspection and maintenance

## 2.3 Production protocol

All $pH_2$ batches were produced in the same manner (the indices in the brackets relate to in Fig. 3):

*Preparation:*

- Set initial state: Close valve A4, connect the generator with the output via A6 ("fill" position)

- Open and set supply of hydrogen to the appropriate pressure

- Connect the storage bottle A8 to the output

- Fill the dewar with $lN_2$ and close the lid to reduce evaporation

- Wait for 20 min and set the flow with the regulators A5


*2. Flushing storage bottle:*

- open valve A4 and wait until the pressure gauge shows 3 bar

- release gas from storage cylinder by connecting the storage bottle to the exhaust via A6 ("venting position")

- repeat the flushing steps three times


*3. Production and storage of pH₂:*

- Set valve A6 to "fill" position

- Wait until the gauge shows the pressure desired

- Close valve A4






*4: Finishing production of pH₂:*

- Close storage bottle (bottle valve)

- Set valve A6 to "vent" position to reduce pressure in the output line

- Disconnect storage bottle from the output (fast connect adapters keeps line closed)

- Set valve A6 to "close" position

- Close $H_2$ supply

## 2.4 Quantification

### Flow quantification

We refrained from including a flow meter in the setup to keep it simple and robust. Instead, we used the time $t^{p,V}$ needed to fill
a cylinder of a given volume $V_0$ to a given pressure $p_{out}$ to measure the average flow rate $f_r$ of $pH_2$ production. To obtain
Standard Liters per Minute (SLM) we used the following equation:

$$f_r = \frac{V_0}{t^{p,V}} \text{ [SLM]} = \frac{P_{out} V_0 T_{norm}}{T_{rt} P_{norm}} \cdot \frac{1}{t^{p,V}} \qquad (2)$$

where $T_{rt}$ is the temperature of the quantification experiment (here: 22 °C) and "norm" stands for standard pressure and
temperature values ($p_{norm} = 10^5$ pascals $\triangleq$ 1.0 bar, $T_{norm} = 273.15$ K) (Nič et al., 2009). The measurement $f_r$ is performed in a
regime where $P_{out}$ in still linear as a function of time ($t^{p,V}$), hence it coincides with an initial flow rate that is usually reported.

### Gas system

A medium-pressure 5 mm NMR tube (522-QPV-8, Wilmad-LabGlass, Vineland, U.S.A.) was used for $pH_2$ quantification
and heavy wall 5 mm NMR tube (Wilmad-LabGlass, 522-PV-9) for experiments with Magnetic Field Cycling (MFC). Each
of these NMR tubes was equipped with input and output gas lines (1/16" PolyTetraFluoroEthylene capillary (PTFE) with
0.023" inner diameter) by glueing to the cap. The other end of these tubes was connected to a custom made valve system. The
pressure in the system was set by changing the reducers of respective gases and back pressure valve in the gas system (P-785,
P-787, Postnova). Inlet gas pressure was regulated to achieve a steady bubbling for the given backpressures of 2.8 bar or
6.9 bar. The valve system is controlled with an Arduino which was linked to the spectrometers software synchronizing the gas
supply, venting of the NMR tube, and data acquisition. Using this gas system we supplied to NMR tube $N_2$ (99.999 %, Air
Liquide), $H_2$ (99.999 %, Air Liquide) or $pH_2$.

### $pH_2$ quantification protocol

The $pH_2$ quantification was performed according to a quantification protocol (schematically shown in Fig. 4).

NMR tube is placed in a 1 T NMR spectrometer (benchtop, SpinSolve Carbon 43 MHz, Magritek, Aachen, Germany) and not
moved during the experiment. To remove air and residual gases from the lines, the setup was flushed with the gas for 3 min at
5 bar input pressure and fully open exhaust. Afterwards, the exhaust line was closed and a 30 s delay was allowed to stabilize
pressure and flow before the NMR acquisition was started. To ensure the constant pressure in the system the gas supply was





kept open during the NMR measurement. Because the NMR signal was not locked during the experiment, the $H_2$ resonance was moved to 0 ppm during post-processing for convenience.

All NMR spectra of gases were acquired with a standard excitation and acquisition of free induction decay pulse sequence (12.6 µs excitation pulse that corresponds to 90° flipping angle, 20 ms acquisition time, 50 kHz spectral width, 0.5 s repetition time, 100 transients for averaging, SpinSolve Expert v3.54, Magritek, Aachen, Germany). Spectra were subjected to 20 Hz exponential apodization and phase-correction. To remove background signals, a spectrum of $N_2$ was acquired also and subtracted from the $rtH_2$ ($H_2$ in thermal equilibrium at room temperature) and $pH_2$ spectra. After that, an automatic baseline correction (MNova v14.1.2, Santiago de Compostela, Spain) was applied to the phased spectrum. The spectral lines of $rtH_2$ and $pH_2$ were integrated within the borders of -15 ppm and +15 ppm giving $S(rtH_2)$ and $S(pH_2)$. And finally the fraction of $pH_2$ $f_{pH_2}$ was calculated:

$$f_{pH_2} = \left(1 - \frac{3}{4}\frac{S(pH_2)}{S(rtH_2)}\right) \cdot 100\ \% \tag{3}$$

Here it is taken into account, that only $oH_2$ contributes to MR signal and $f_{pH_2} = \frac{1}{4}$ at room temperature (Green et al., 2012).

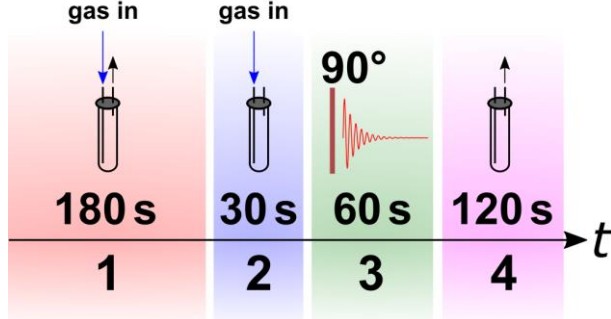

**Figure 4: Scheme of $pH_2$ quantification protocol.** The NMR tube is flushed with $N_2$, $pH_2$ or $rtH_2$ gas for 180 s before the exhaust is closed. A rest time of 30 s is allowed for the system to settle down. Finally, the NMR spectra are acquired, before the gas is released.

## 2.5 SABRE experiment

**Sample preparation.** The sample solution contained 3 mmol/L iridium N-heterocyclic carbene complex [Ir(COD)(IMes)Cl], where COD = 1,5-cyclooctadiene, Imes = 1,3-bis(2,4,6-trimethylphenyl) imidazol-2-ylidene (Cowley et al., 2011) and 26 mmol/L nicotinamide (CAS 98-92-0, Sigma-Aldrich) in methanol-$d_4$ 99.8 % (Deutero GmbH). To activate the catalyst, $H_2$ was flushed through the sample at 6.9 bar for 5 minutes before the magnetic field cycling experiments.

**Magnetic field cycling experiment.** The NMR spectrometer was equipped with an in-house built MFC setup that will be described elsewhere. Shuttling time from the observation point to the sweet spot of the electromagnet was 0.2 seconds. The used electromagnet allowed a magnetic field variation in the range of -20 mT to 20 mT with a magnetic field homogeneity of 0.06 % in 2 cm. The same gas system as described above was used for the MFC SABRE experiments. The only modification



was that a hollow optical fibre (Molex, part. num. 106815-0026, 250 mm internal diameter, 360 mm outer diameter) was glued to the end of the PTFE capillaries to reduce magnetic field distortions. All magnetic field-cycling SABRE experiments were carried out according to the protocol on Fig. 5.


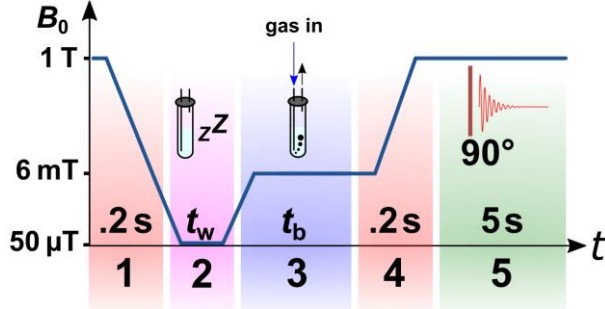

**Figure 5: Scheme of $^1$H magnetic field cycling SABRE experiment.** *Stage 1*: shuttling of the sample to the polarization coil. *Stage 2:* relaxation of the sample at Earth's magnetic field for $t_w = 10$ s. *Stage 3*: switching on the electromagnet with a magnetic field $B_p = 6$ mT and starting bubbling with $p$H$_2$ enriched gas at pressure $P = 6.9$ bar or 2.8 bar for $t_b = 30$ s. *Stage 4*: shuttling of the sample to the bore of the

NMR spectrometer in 0.2 seconds and turning off the electromagnet. *Stage 5:* after 90º excitation, acquiring of the $^1$H-NMR spectrum.

## 3 Results

### 3.1 PHG design

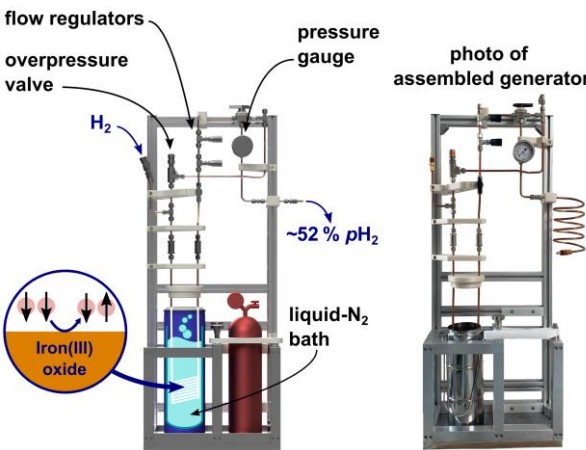

**Figure 6: a) Rendering of the PHG (left) and a photo of the final build (right).** The design of the PHG is open-source, simple and uses off-the-shelf as well as 3D-printed parts.





A PHG fulfilling the initial design requirements was successfully constructed (Fig. 6). Most parts were either commercially available, 3D printed or simple to construct on-site. The holders for the bottles and a bottom plate were the sole part prepared

by a mechanical workshop. All parts were rated for more than 100 bars and no $H_2$ leaks were detected at 50 bars of $H_2$ using an $H_2$ detector (GasBadge Pro H2, Industrial Scientific, Pittsburgh, U.S.A.). Inspection and operation were facilitated by easy access and open construction design. The total cost was below 3,000 € (Appendix A).

We deliberately abstained from including a flow meter into the setup to keep the cost low and increase the robustness. Instead,

we monitored the pressure $P_{out}$ in the storage cylinder and calculated the flow rate (Fig. 7a). The expected increase in pressure and decrease in the flow rate of $pH_2$ was observed. The flow rate is an important parameter since it affects the collisions of $H_2$ with the catalyst in the *ortho-para* conversion unit (Fig. 3, A3). At optimal parameters, $lN_2$ based PHG can provide $f_{pH_2} \cong$ 52 % (Fig. 2,7b). These collisions are responsible for the fast *ortho-para* conversion. If the flow rate is too fast, the gas will not have enough time to reach *ortho-para* thermal equilibrium while passing through the unit. Hence the $pH_2$ fraction will be

reduced.

Thus, to find optimal performance conditions of the PHG we quantified $f_{pH_2}$ as a function of the flow rate (Fig. 7c) set by the needle valves (Fig.3, A5). At the given settings of $P_{in}$ = 20 bar and $P_{out}$ = 10 bar, $f_{pH_2} \approx$ 51.7 % was found for a flow up to $f_r$ = 2 SLM. For larger flow rates, the enrichment dropped significantly. Given this data, and to allow for some variation, we chose a standard operating flow of $\approx$ 0.9 SLM. This flow rate was fast enough for convenient ad-hoc $pH_2$ production. For

example, l L of 49 bar $pH_2$ with $f_{pH_2}$ = (51.7 ± 0.76) % were produced in 29 min ($P_{in}$ = 49 bar, initial flow rate of 2.9 SLM, Fig. 7a).

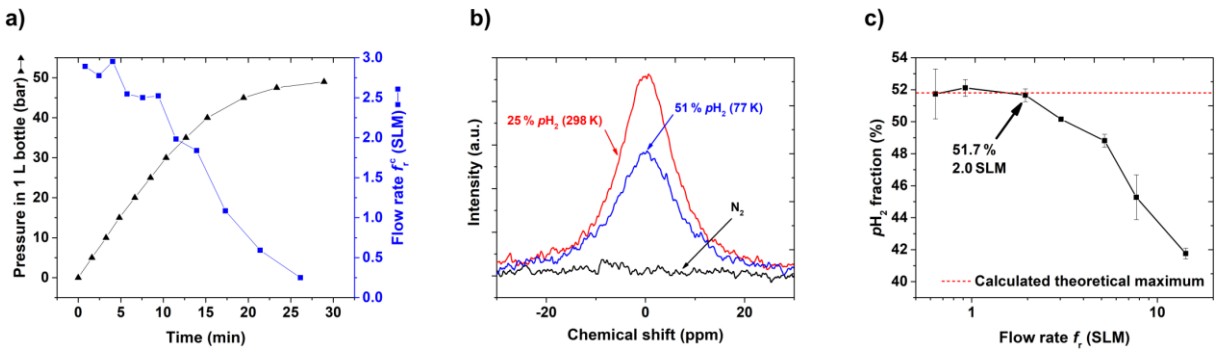

**Figure 7: PHG operating parameters and NMR spectra:** (a) Pressure $P_{out}$ and calculated flow rate $f_r^c = P'_{out} \cdot \frac{V_0 T_{norm}}{T_{rt} P_{norm}}$ as a function of time for input pressure $P_{in}$ = 50 bar and $V_0$ = 1 L, (b) $^1$H NMR spectra of rt$H_2$, $pH_2$ and $N_2$ to quantify $f_{pH2}$, and (c), $f_{pH2}$ as a function of $f_r$

(eq. 3). For the latter, the para-enrichment was found to be constant up to a flow rate of $f_r$ = 2 SLM (for $P_{in}$ = 20 bar, $P_{out}$ = 10 bar)



### 3.2 The precision of $p\text{H}_2$ production, quantification and lifetime

To test the reproducibility of the quantification method, $f_{p\text{H}_2}$ of a single batch was quantified 5 times in a row (including

venting, flushing, and filling of the tube). The average $f_{p\text{H}_2}$ was found to be (51.5 ± 0.36) %, corresponding to a coefficient of

variation (CV) of 0.7 % (Fig. 8).

To access the reproducibility of the entire production process, four $p\text{H}_2$ batches were produced on different days. An average

$f_{p\text{H}_2}$ of (51.6 ± 0.88) % was observed (CV = 1.7 %) (Fig. 8).

For evaluating the lifetime of $p\text{H}_2$ in the 2 L cylinder, a 10 bar $p\text{H}_2$ batch was produced ($P_{\text{in}}$ = 20 bar, $f_{\text{r}}$ = 0.9 SLM). Over

22 days, five samples were taken from the batch and $f_{p\text{H}_2}$ was quantified. An exponential decay function was fitted to the data

and yielded a constant of 35.5 ± 1.48 days (Fig. 9).

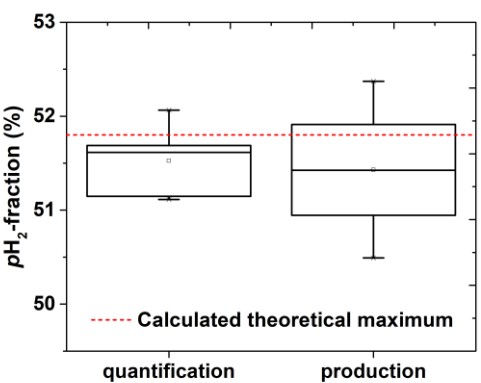

**Figure 8: $p\text{H}_2$ quantification and production reproducibility chart.** The left boxplot shows the precision of quantification method. The

$p\text{H}_2$ quantification protocol was repeated five times with the same $p\text{H}_2$ batch. The obtained $p\text{H}_2$ fraction was (51.5 ± 0.36) % that gives us

an impression of quantification precision. The right boxplot shows the reproducibility of the production. The production of $p\text{H}_2$ and

quantification protocols were repeated once on four different days. The obtained $p\text{H}_2$ fraction here was (51.6 ± 0.88) %; the error value

includes production and quantification errors. PHG parameters of $p\text{H}_2$ preparation: 20 bar inlet pressure, 10 bar final pressure in the storage

cylinder of and an of 0.9 SLM average flow rate. All errors are given by the standard deviation.



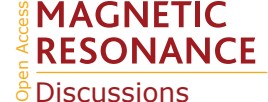

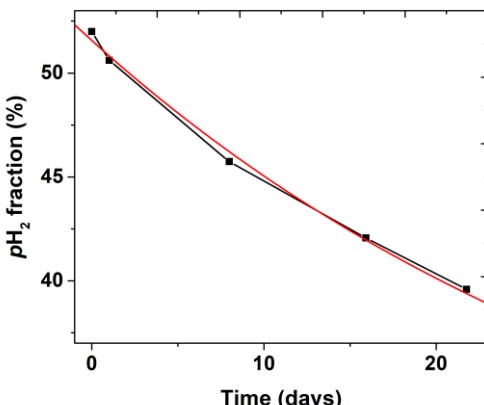

**Figure 9: $p$H$_2$ lifetime in a 1 L aluminium cylinder.** The data are fitted with the exponential decay function: $A_1 \cdot \exp(-t/\tau) + y_0$ with $y_0$ fixed to 25, $A_1 = 26.6 \pm 0.29$ and relaxation decay time $\tau = (35.5 \pm 1.48)$ days

### 3.4 Application: $^1$H-low-field SABRE at different $p$H$_2$ pressures

The presented setup was designed to allow for pressures up to 50 bar. High pressures are beneficial for hyperpolarization
because the concentration of $p$H$_2$ in solution increases with pressure. Low concentration of $p$H$_2$ is often the limiting factor of the hyperpolarization yield (polarization level × concentration of polarized species). To demonstrate the effect, we polarized nicotine amide by SABRE and magnetic field-cycling (scheme at Fig. 5) at two different $p$H$_2$ pressures: 2.8 and 6.9 bar (Fig. 10). Strong polarization was observed on $^1$H resonances of nicotine amide and hydrogen in solution and increased at higher pressure. A 2.5 fold increase in pressure yielded a 2.3 fold increase of nicotine amide polarization.

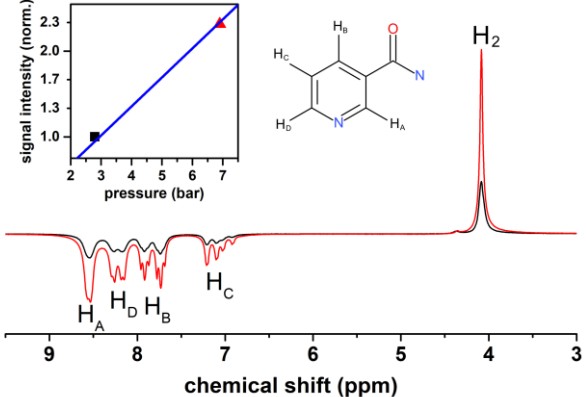


**Figure 10: $^1$H-SABRE spectra of nicotine amide and H$_2$ at 6.9 bar and 2.8 bar $p$H$_2$ pressure.** (Insert) The signal intensity of nicotine amide vs. pressure with a linear fit (blue line). Nicotine amide structure is added for convenience.



## 4 Discussion

**Design.** The design of the presented PHG is simple and compact without compromising on performance and safety. The PHG is small and portable (although a heavy bottom plate was added to add stability). Since there are no electrical components, it can be placed indoor as well as outdoor and does not require any electrical power supply. Note, electric components can be an ignition source which may lead to an explosion in case of a hydrogen leak.

For the framework, mostly, off-the-shelf parts were used. More complex geometries as e.g. holders for valves or gauges were
3D-printed. They have individual shapes and dimensions and manufacturing in a workshop might lead to high costs and long production lead times. 3D-printing turned out to be a versatile manufacturing method enabling fast prototyping, complex shapes, and low-cost for one-off productions. The design of the PHG, all 3D-models (STL-files, Standard Triangulation Language, and CAD-files, Computer-Aided Design) are provided enabling other groups to adjust the parts to their individual needs (Ellermann, 2020b).

Choosing a small 2 L dewar keeps the design compact and the running costs low since less than 2 L of liquid-nitrogen were required to prepare 1 L of $pH_2$ at 50 bar. In combination with a short cooling time, the setup is perfectly suited for on-site $pH_2$ production in a hyperpolarization lab.

**Costs.** The final cost of the PHG incl. the hydrogen sensor is 2,988 € incl. VAT (19 %). If a hydrogen sensor is already available in the lab, the overall costs for the PHG can be pushed down to less than 2,500 € incl. VAT (19 %). A complete set
including the PHG, a hydrogen sensor, hydrogen/nitrogen gas as well as a variety of essential tools costs about 3,700 € incl. VAT (19 %).

**Safety.** All parts which are in contact with pressurized gas are rated to at least 100 bar. However, we fixed the operation pressure to 50 bar to get a generous safety margin of 100 %. Additionally, the design of the PHG incorporates a gas path which also enables safe ventilation of storage bottle. The design and the choice of parts also consider potential handling errors. For
example, the output connectors are closed for pressures up to 17 bar when they are disconnected, i.e. the storage bottle is disconnected. Thus, no contact between the air in the room and hydrogen in the PHG occurred. Furthermore, we included a handheld hydrogen sensor that can measure hydrogen concentrations in the parts per million (ppm) regime. The sensor should be always turned on during operation and will indicate potential leakages of $H_2$ gas.

The setup includes low-temperature cryogens as liquid nitrogen. To prevent the spilling of liquid nitrogen, the dewar is
restrained by the copper tubing inside and the surrounding metal frame. Moreover, a lid covers the liquid nitrogen bath that also reduces the evaporation rate of the cryogen. Since the flask only holds around 2 L of cryogen, the amount of liquid nitrogen that has to be handled is greatly reduced.

Note that PHG should be placed in a non-public lockable enclosure or room with sufficient ventilation and only instructed personnel should operate it.




Our presented safety concept is economical and practical without sacrificing any safety measures. Nevertheless, after setting up the PHG, it should be tested for leakage with inert gas or nitrogen. The part-list also contains a leak detection spray and sensor.

**Performance** The enrichment achieved here, e.g. $f_{pH_2} = 51.6 \pm 0.88$ % for $P_{in} = 20$ bar, $f_r = 0.9$ SLM, was close to the

maximum of 51.8 % conditioned by the boiling point of $lN_2$, and somewhat higher than reported elsewhere $f_{pH_2} = 50$ % (Barskiy et al., 2016a, 2016b; Shchepin et al., 2016). Determining the enrichment as a function of flow allowed us to choose an optimal flow of 0.9 SLM for $P_{in} = 20$ bar: this rate is sufficient e.g. to fill 1 L bottles to 10 bar in 10 min. The central design criterium of high pressure was successfully met as 1 L of 49 bar $pH_2$ were produced in 28 min ($P_{in} = 50$ bar). We demonstrated that an increase of $pH_2$ pressure can give a proportional increase in polarization (Fig. 10). Obviously, this approach is limited

as soon as the hyperpolarization yield is no longer determined by the availability of $pH_2$ and cannot provide polarization above 33% (Korchak et al., 2018).

**$pH_2$ quantification and production reliability** The automatic quantification process features a CV of 0.7 %; the $pH_2$ production and quantification together feature a CV of 1.7 % (Fig. 8). Both values are suited for routine $pH_2$ quality control without a need for an expensive high-field NMR system. The automatization certainly helps to make the process more reliable

but is not necessary. Feng et al. used the same quantification approach and reported precision of 1-3 % for quantification (2012). NMR is a convenient method for $pH_2$ quantification, but optical methods may be used, too (Parrott et al., 2019).

**The lifetime in aluminium cylinder** Feng et al. reported a lifetime in aluminium tanks of $(63.7 \pm 8.3)$ days and about 2 % points loss of $f_{pH_2}$ per week (2012). With ~120 days of lifetime, Hövener et al. reported even longer values (2013). We found here a shorter lifetime of $(35.5 \pm 1.48)$ days in our 2 L aluminium storage bottle. Note, that we did not vacuum our cylinder

that can increase the lifetime. Still, the lifetime is sufficiently long to produce $pH_2$ once a week; the lifetime of $(35.5 \pm 1.48)$ days corresponds to about 5.5 % points loss of $f_{pH_2}$ per week.

## 5 Conclusion

The presented PHG provides $f_{pH_2} \approx 52$ % at a high pressure of 50 bar reliably ($CV = 1.7$ %) that provides about 1/3 of the polarization achieved with $f_{pH_2} \approx 100$ %. Because the device provides high-pressure $pH_2$, however, this effect can be partially

compensated in the PHIP/SABRE experiment. A new, automated quantification routine at 1 T benchtop NMR proved to be reliable and simple ($CV = 0.7$ %). The design of PHG is straightforward, easy to manufacture with openly available blueprints and at a cost of less than 3,000 €. The device may facilitate further research on the promising method of parahydrogen-based hyperpolarization.





**6 Appendices**

**Table A1. Price list of all needed components for high-pressure PHG.** Full list of items required for construction of portable liquid nitrogen para-hydrogen generator (without cart) with 6 mm copper connection tubes. Here following format for item names is used: {Name [Company], (Specifications), Article number}. Given prices include 19 % VAT.

| № | Name | Amount | Price, incl.VAT (Euro/unit) | Price, incl. VAT (Euro) |
|---|------|--------|------------------------------|--------------------------|
| **Gases and cylinders (148.00 €)** | | | | |
| 1 | Hydrogen cylinder [Airliquid], (Gas, ALPHAGAZ1, S10-1.8 m3), P0231S10R2A001 | 1 | 30.00 | 30.00 |
| 2 | Nitrogen cylinder [Airliquid], (Gas, ALPHAGAZ1, S10-1.8 m3), P0271S10R2A001 | 1 | 40.00 | 40.00 |
| 3 | Aluminium cylinder [Luxfer], (Cylinder, $H_2$ valve 28,8/21,8 LH), A6341Q | 1 | 78.00 | 78.00 |
| **Regulators, gauges and valves (1578.92 €)** | | | | |
| 4 | Hydrogen cylinder pressure regulator [AirLiquide], (EUROJET 200/50, DIN 477-1, 6 mm tube outlet), Eurojet_125607 | 1 | 279.00 | 279.00 |
| 5 | Nitrogen cylinder pressure regulator [Kayser], (in 200 bar, out 0-20 bar, DIN 477-1, 1/4" mm male ISO parallel thread), CK1302 | 1 | 128.00 | 128.00 |
| 6 | Stainless Steel 1-Piece 40G Series Ball Valve [Swagelok], 0.6 Cv, 6 mm Tube Fitting | 1 | 124.12 | 124.12 |
| 7 | Stainless Steel High-Pressure Proportional Relief Valve [Swagelok], 6 mm Tube Fitting | 1 | 253.83 | 253.83 |
| 8 | Purple Spring for Proportional Relief Valve [Swagelok], 750 to 1500 psig (51.7 to 103 bar) | 1 | 6.90 | 6.90 |
| 9 | Stainless Steel 1-Piece 40G Series 3-Way Ball Valve [Swagelok], 0.90 Cv, 6 mm Tube Fitting | 1 | 191.00 | 191.00 |
| 10 | Stainless Steel Low Flow Metering Valve [Swagelok], 6 mm Tube Fitting, Vernier Handle | 2 | 242.52 | 485.04 |
| 11 | Gauge up to 100 bar [Swagelok], (6 mm tube fitting) | 1 | 111.03 | 111.03 |
| **Tube and fittings (346.15 €)** | | | | |
| 12 | Copper tube, CUR6X1R Kupferrohr 6x1 mm, R 220, rolled good, soft, 1m [Landefeld] | 5 | 11.29 | 56.45 |



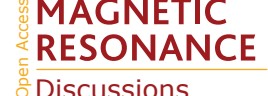

| 13 | Brass Instrumentation Quick Connect Stem with Valve [Swagelok], 0.2 Cv, 6 mm Tube Fitting | 2 | 27.01 | 54.03 |
|---|---|---|---|---|
| 14 | Brass Instrumentation Quick Connect Body [Swagelok], 0.2 Cv, 6 mm Bulkhead Tube Fitting | 2 | 54.43 | 108.86 |
| 15 | Stainless Steel Tube Fitting [Swagelok], Union, 6 mm Tube OD | 1 | 17.91 | 17.91 |
| 16 | Stainless Steel Swagelok Tube Fitting, Union Tee, 6 mm Tube OD | 3 | 36.30 | 108.90 |
| **Rest (Catalyst, H₂ detector, Dewar,…, 1278.20 €)** | | | | |
| 17 | Iron(III) oxide [Merck] (catalyst grade, 30-50 mesh), 371254-50G | 1 | 152.00 | 152.00 |
| 18 | Hydrogen detector [Industrial Scientific], (GasBadge Pro H2, 0–2000 ppm), 18100060-C | 1 | 499.00 | 499.00 |
| 19 | Stainless steel liquid nitrogen Dewar flask [Isotherm], (DSS 2000, 2L, h=305, D=114), 2103 | 1 | 413.00 | 413.00 |
| 20 | Stainless Steel In-Line Particulate Filter [Swagelok], 6 mm Tube Fitting, 90 Micron Pore Size | 2 | 107.10 | 214.20 |
| **Gas exhaust line (59.80 €)** | | | | |
| 21 | Polyamid tube [AS-Drucklufttechnik GmbH], (6x3 mm, 1 m), PA 6X3 BLAU-25 | 5 | 1.36 | 6.80 |
| 22 | Multiple plug connection [AS-Drucklufttechnik GmbH], (1x8mm, 4x6mm), IQSQ 8060 | 1 | 16.00 | 16.00 |
| 23 | Reduction plug nipple [AS-Drucklufttechnik GmbH], (8mm to 6mm), IQSG 80H60 | 1 | 5.90 | 5.90 |
| 24 | Check valve [AS-Drucklufttechnik GmbH], (6mm, 0.2 opening <0.2bar), HIQS 60 | 1 | 31.10 | 31.10 |
| **Tools (125.00 €)** | | | | |
| 25 | Clip [AS-Drucklufttechnik GmbH], (0 - 14 mm), SAS 14 | 1 | 9.2 | 9.20 |
| 26 | Leak Detection Spray, [HM INDUSTRIESERVICE GMBH], 291-1252 | 1 | 15.1 | 15.10 |
| 27 | Adjustable wrench RF 300 [Proxxon], (max 34 mm), 23994 | 1 | 26 | 26.00 |
| 28 | Open ended wrench set | 1 | 16.2 | 16.20 |
| 29 | Pipe cutter [Landefeld], (3 - 30 mm), 4333097000773 | 1 | 57.5 | 57.50 |
| 30 | PTFE thread seal tape | 1 | 1.00 | 1.00 |
| **Frame parts (132.17 €)** | | | | |





| 31 | Rexroth profile 2 m each (2x 980 mm, 5x340 mm) | 2 | 38.28 | 76.56 |
| 32 | Rexroth *Nutenstein* M4/M5 | 14 | 1.12 | 15.66 |
| 33 | Filament for 3D-printer: Ultimaker PLA *Perlweiß* 2,85 mm 750 g | 1 | 39.95 | 39.95 |
| 34 | In-house built metal parts (e.g. from university workshop) | 0 | 0 | 0 |
| 35 | Screws M4/M5 | 0 | 0 | 0 |
| | Total price for the basic equipment incl. Hydrogen sensor, excl. VAT | | | 2511.12* |
| | **Total price for the basic equipment, incl. 19 % VAT** | | | **2988.24*** |
| | Total price for the complete set incl. tools and gases, excl. VAT | | | 3082.55* |
| | Total price for the complete set incl. tools and gases, incl. 19 % VAT | | | 3668.24* |


**\*** Overall costs may vary due to change of prices, change of VAT rate, and due to costs which may arise for custom parts (e.g. material or labour costs from the facility's workshop)

## 7 Code availability

Software for automatic signal recording and gas control will be available from figshare.com (DOI: http://dx.doi.org/10.6084/m9.figshare.13176830) (Ellermann, 2020a) and via git (Ellermann, 2020b).


## 8 Data availability

All experimental data (SpinSolve [1]H NMR spectra of $H_2$) and blueprints for the PHG will be available from figshare.com (DOI: http://dx.doi.org/10.6084/m9.figshare.13176830) (Ellermann, 2020a). Additionally, all blueprints are also accessible via git (Ellermann, 2020b).

**9 Team list**

Frowin Ellermann (https://orcid.org/0000-0001-6446-6641)

Andrey N. Pravdivtsev (https://orcid.org/0000-0002-8763-617X)

Jan-Bernd Hövener (https://orcid.org/0000-0001-7255-7252)



## 10 Author contribution

Data curation, investigation, formal analysis, software development (here programming of macros), validation, visualization and writing of the original draft was done by FE. ANP and JBH contributed equally to conceptualization, supervision and reviewing the manuscript.

## 11 Competing interests

There are no competing interests to declare.

## 12 Disclaimer

## 13 Financial Support

We acknowledge support by the Emmy Noether Program "metabolic and molecular MR" (HO 4604/2-2), the research training circle "materials for brain" (GRK 2154/1-2019), DFG - RFBR grant (HO 4604/3-1, № 19-53-12013), the German Federal Ministry of Education and Research (BMBF) within the framework of the e:Med research and funding concept (01ZX1915C), 
Cluster of Excellence "precision medicine in inflammation" (PMI 1267). Kiel University and the Medical Faculty are acknowledged for supporting the Molecular Imaging North Competence Center (MOIN CC) as a core facility for imaging in vivo. MOIN CC was founded by a grant from the European Regional Development Fund (ERDF) and the Zukunftsprogramm Wirtschaft of Schleswig-Holstein (Project no. 122-09-053).

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
