# Peer review of "Open-source, partially 3D-printed, high-pressure (50 bar) liquidnitrogen-cooled parahydrogen generator"

_Magnetic Resonance, 2020_

## Short Comment (SC1) · 15 Nov 2020

Given that parahydrogen-based research area keeps growing rapidly, I believe that this work will be of intetrest and use for many researchers, and particularly to those who consider entering the field. Therefore, in my opinion the paper should certainly be published.

Provided below are some suggestions for the authors which may further improve the manuscript.

1) From the title, I had an impression that the generator or at least its key parts were

3D-printed, which, as I found later, was not the case. I believe it may be a good idea to refine the paper title. 2) As always, I'm advocating the spelling of "parahydrogen" and "orthohydrogen" as single words without a dash, which I believe is the only correct way to spell them (cf. paratrooper, parabola, orthophosphoric acid; also in dictionaries, e.g., https://www.thefreedictionary.com/parahydrogen). 3) The word "allotrope" in the reference to parahydrogen is acceptable in the historic context, but in fact is incorrect. By definition, allotropy is the existence of a chemical element in two or more forms, which may differ in the arrangement of atoms in crystalline solids or in the occurrence of molecules that contain different numbers of atoms (e.g., graphite, charcoal, diamond, fullerenes). Parahydrogen is this not an allotropic form of $H_2$ but rather its nuclear spin isomer. 4) line 40, naphthalene derivative (Stevanato et al., 2015) cannot be classified as spin isomer as it is not symmetric, so that the generalized Pauli principle is not applicable to it. 5) Line 39, water (Mammoli et al., 2015; Meier et al., 2015) was not enriched in the Mammoli paper experiments, or at least could not be extracted to RT. A better reference to spin isomers of free water would be to the molecular beam separation experiments (e.g., doi: 10.1126/science.1200433 or 10.1021/acs.jpca.9b04294). 6) There are a couple of very recent papers describing different parahydrogen generator designs which the authors may wish to cite, doi: 10.1021/acs.analchem.0c03358 doi: 10.1016/j.jmr.2020.106869

---

## Referee Comment (RC1) · Anonymous Referee #1 · 17 Nov 2020

The authors describe construction and operation of high-pressure (50 bar) liquid-nitrogen-based parahydrogen generator (PHG) built from readily available components (total price <3k Euros). They demonstrate applications of PHG to SABRE hyperpolarization, quantify the precision of pH2 production and measure pH2 lifetime in a 1L aluminum cylinder. The results are overall well presented and are definitely useful for new researchers entering the field of PHIP/SABRE. The paper can be published as is subject to minor corrections. In particular,

Page 2, line 46: '0Z axis'? Should it be Z axis? Page 4, line 84: 'don't require much in terms of service' - poor word choice. Page 4, line 90: 'were performed ... - and still are'

- poor word choice. Page 4, line 95: 'way beyond' - poor word choice. Page 5, line 113: 'Iron (III) oxide' - should be hydrated iron oxide or FeO(OH) Page 8, line 183: should 'norm' stand for normal pressure and temperature (not standard)? See, for example, https://www.engineeringtoolbox.com/stp-standard-ntp-normal-air-d_772.html Page 10, line 227: '250 mm internal diameter, 360 mm outer diameter' - numbers seem too big, please check. Page 11, line 253: 'These collisions'? Page 11, line 259: 'ad-hoc' is used improperly in this sentence. Pge 12, line 269 and 273: errors and mean values should have the same number of significant figures, i.e., either 51.50(36) or 51.5(4)

---

## Referee Comment (RC2) · James Eills (Referee) · 22 Nov 2020

This work describes the construction and operation of a high-pressure parahydrogen generator, and the authors list in detail the required equipment, costs, safety considerations, and provide a discussion of the advantages/disadvantages of this system. Overall this is a nice addition to the literature, and I expect this piece of work will prove to be a useful tool for researchers wanting to bring PHIP+SABRE experiments to their laboratory, or upgrade their current equipment at low cost. This work is especially useful as the system is capable of operating at 100 bar $H_2$ pressure, which exceeds the typical gas pressures reported in PHIP/SABRE experiments; this will likely be an

important step forward for the field in the quest for ever-higher polarization levels and reduced reaction times.

I support publication of this paper after some minor corrections, which I hope the authors will find helpful:

Title: "3D-printed" should be removed from the title or clarified as "partially 3D printed" (or similar), as this is misleading in its current form.

Line 40: The authors might choose to also mention long-lived spin isomers in methyl rotors, e.g. doi: 10.1021/ja410432f

Line 113: Can the authors make any comment about how well the cotton wool works in preventing iron oxide flow through the system, or state how much was used for effective filtering?

Line 141: It would be helpful to briefly state how inductive and static spark charges are avoided.

Line 265: I'm a little bit confused about the reporting of $P_{out}$, doesn't this value vary in time? Does $P_{in}$ describe the gas pressure in the catalyst region? Isn't this the only relevant pressure when considering conversion as a function of flow rate?

Lines 343-344: But the quantification *is* performed on a high-field NMR system. Better to say "We have shown that routine $pH_2$ quality control can be performed with a low-cost benchtop NMR system", or similar..

Lines 347-351: I don't understand the value "points loss of $f_{pH_2}$ per week". A relaxation time is given to describe the exponential process and this should be sufficient to understand the $f_{pH_2}$ change each week. Unless I've misunderstood, I suggest simply removing this.

Line 348: "With ∼120 days of lifetime, Hövener et al. reported even longer values (2013)." The phrasing of this sentence is a little strange.

Line 349: It would be helpful to explain why vacuuming the cylinder should lead to a longer $pH_2$ lifetime (presumably due to removal of paramagnetic oxygen molecules).

Figure 7: It would be informative to include an additional plot showing para-enrichment level as a function of pressure ($P_{in}$) for a 2 SLM flow rate. This is a suggestion and I

do not insist on this for publication.

In another interactive comment, Prof. Igor Koptyug has provided two references to other parahydrogen generators described in the literature. I agree that this closely-related work should be cited.

---

## Referee Comment (RC3) · Eleonora Cavallari (Referee) · 23 Nov 2020

Recent developments in hyperpolarization techniques allow dramatic increases in sensitivity of orders of magnitude that open new perspectives both in preclinical and in clinical application. The possibility of building a very economical high pressure parahydrogen producer is certainly useful for the spread of this technique.

Although 50% para enrichment precludes the possibility of exploiting this generator for in vivo preclinical applications, as demonstrated by the authors it will be very useful for fundamental research.

[Figure]
In this contribution, a well-constructed and well written piece of work, the authors describe the design and construction of a high-pressure low cost parahydrogen generator, also keeping in mind the safety aspects.

The work is undoubtedly interesting, carefully done and nicely presented. In my opinion this paper should be published after some minor revisions.

- I find the title misleading, reading it I thought the authors provided a printable 3D-model of the parahydrogen generator, therefore I suggest to modify it.

- The cotton wool filters used to protect the system from catalyst contamination it could be better described. It would be interesting to know the life time if the cylinder had been vacuumed or even washed with H2/pH2 before being filled with pH2, in order to verify that the value obtained using this generator is comparable with those reported previously.

- I don't understand the meaning of "points loss of f_pH2". Is this value related in some way to the polarization obtainable (1/3) with respect to f_pH2 $\approx$ 100 %? The authors should rephrase the sentence to clarify this concept and/or add the formula that defines it.

- In general, I agree with all the comments proposed by the other reviewers.

---

## Author Comment (AC1) · 11 Dec 2020

Dear Igor Koptyug,

We thank you for your valuable comments which significantly improved our manuscript. We appreciate that you used the open discussion section of Magnetic Resonance journal. Hereby, we want to provide a point-by-point response to your post.

Your comment: "From the title, I had an impression that the generator or at least its essential parts were 3D-printed, which, as I found later, was not the case. I believe it may be a good idea to refine the paper title."

Answer: We understand that the title could be misleading and to avoid it the new title reads as "Open-source, partially 3D-printed, high-pressure (50 bar) liquid-nitrogen-cooled parahydrogen generator"

Your comment: "As always, I'm advocating the spelling of "parahydrogen" and "orthohydrogen" as single words without a dash, which I believe is the only correct way to spell them (cf. paratrooper, parabola, orthophosphoric acid; also in dictionaries, e.g.,https://www.thefreedictionary.com/parahydrogen)."

Answer: We understand your point and removed the dash. Additionally, we stick to the non-italic form.

"para-hydrogen" -> "parahydrogen" throughout the text

Your comment: "The word "allotrope" in the reference to parahydrogen is acceptable in the historic context, but in fact is incorrect. By definition, allotropy is the existence of a chemical element in two or more forms, which may differ in the arrangement of atoms in crystalline solids or in the occurrence of molecules that contain different numbers of atoms (e.g., graphite, charcoal, diamond, fullerenes). Parahydrogen is this not an allotropic form of H2 but rather its nuclear spinisomer."

Answer: We changed the word allotrope/allotropic to "nuclear spinisomer" except for the citation and the text referring to it.

Old text (e.g.): "The spin of hydrogen nuclei (proton) is the origin of the two allotropic forms or two spin isomers of hydrogen."

New text: "The spin of hydrogen nuclei (of protons) is the origin of the two nuclear spinisomer forms of dihydrogen."

Your comment: "line 40, naphthalene derivative (Stevanato et al., 2015) cannot be classified as spin isomer as it is not symmetric, so that the generalized Pauli principle is not applicable to it."

and

"Line 39, water (Mammoli et al., 2015; Meier et al., 2015) was not enriched in the Mammoli paper experiments, or at least could not be extracted to RT. A better reference to spin isomers of free water would be to the molecular beam separation experiments (e.g., doi: 10.1126/science.1200433 or 10.1021/acs.jpca.9b04294)."

Answer: We rephrased the sentences and added proposed references.

Old text (e.g.): "Hydrogen is not the only compound that has stable or long-lived spin isomers at room temperature (rt) there are many examples: deuterium (Knopp et al., 2003), water (Mammoli et al., 2015; Meier et al., 2015), ethylene (Zhivonitko et al., 2013), and even naphthalene derivative (Stevanato et al., 2015)."

New text: "Hydrogen is not the only compound that has stable or long-lived spinisomers at room temperature (rt) there are many examples: deuterium (Knopp et al., 2003), water (Kravchuk et al., 2011; Vermette et al., 2019) , ethylene (Zhivonitko et al., 2013) and methyl groups (Meier et al., 2013). Although some molecules are not symmetric and cannot be extracted at room temperature, they possess long-lived spin states of minutes (Pileio et al., 2008) and hours (Stevanato et al., 2015)."

Your comment: "There are a couple of very recent papers describing different parahydrogen gener-ator designs which the authors may wish to cite, doi: 10.1021/acs.analchem.0c03358doi: 10.1016/j.jmr.2020.106869"

Answer: These papers we not yet published when we finished our manuscript. Nevertheless, we included both generators to our overview table (Table 1). The updated table is attached to this comment.

We want to thank you again for sharing your thoughts in the open discussion. We picked this journal since we like the open access and open discussion approach.

With kind regards, Frowin Ellermann and Jan-Bernd Hövener

**Table 1: Performance comparison of several PHGs:** (1) Bruker PHG 90, (2) dual-stage cryostats (DSC) (Hövener et al., 2013), (3) a pulsed PHG (Feng et al., 2012), (4) HyperSpin-PHG (Meier et al., 2019), (5) Automated PHG (Birchall et al., 2020), (6) He-dewar PHG (Du et al., 2020), (7) U-shape PHG (Kiryutin et al., 2017), (8) economical PHG (Jeong et al., 2018), (9) glass-trap PHG (Gamliel et al., 2010) and (10) in house designed and built PHG (this work). Given prices include all connectors, cylinders, and 19 % VAT. $lN_2$ stands for liquid nitrogen and "cc-He" for closed-cycle He compressor.

| | Name | Operating temperature (K) [method] | $f_{pH_2}$ (%) | Initial flow rate (SLM) | Max. pressure (bar) | Price (€) |
|---|---|---|---|---|---|---|
| 1 | Bruker BPHG 90 | 36-40 [cc-He] | 85-92 | ≤ 0.2 | 10 | 100,000–150,000 |
| 2 | DSC (Hövener et al., 2013) | 25 [cc-He] | 98 ± 2 | 4 | 50 | 37,000 |
| 3 | Pulsed PHG (Feng et al., 2012) | 15 [cc-He] | 98 | 0.9 | 20 | N.A. |
| 4 | HyperSpin-PHG (Meier et al., 2019) | 20-77 [cc-He] | N.A.[a] | N.A. | Min. 10 | N.A. |
| 5 | Automated PHG (Birchall et al., 2020) | 40 [cc-He] | ~87 | 0.15 | 33.8 | <25,000 |
| 6 | He-dewar PHG (Du et al., 2020) | 30 [He] | 97.3 ± 1.9 | ~0.3 | 4.5 | N.A. |
| 7 | U-shape PHG (Kiryutin et al., 2017) | 77 [$lN_2$] | ~50 | 0.36[b] | Min. 3 | N.A. |
| 8 | Economical PHG (Jeong et al., 2018) | 77 [$lN_2$] | ~50 | N.A. | N.A. | N.A. |
| 9 | Glass-trap PHG (Gamliel et al., 2010) | 77 [$lN_2$] | 46.3 ± 1.3 | 0.0025[c] | ~1 | N.A. |
| 10 | This work | 77 [$lN_2$] | 51.6 ± 0.9 | 2.0[d] | 50[e] | 2,988[f] |

**Fig. 1.**

---

## Author Comment (AC2) · 11 Dec 2020

Dear referee,

We thank you for your valuable comments. We appreciate that you've read the manuscript thoroughly. Overall, we implemented all your comments, and in the following, we want to provide a point-by-point response.

**Your comment:** "In particular,Page 2, line 46: '0Z axis'? Should it be Z axis?"

**Answer:** As proposed, we changed "0Z axis" to more common "Z axis" in the text

**Your comment:** "Page 4, line 84: 'don't require much in terms of service' - poor word choice."
**Answer:** We agree and modified the sentence. Actually, the point here is that these setups don't require the supply cryogens.
**Old text:** "These setups work reliably and don't require much in terms of service (e.g. no liquid cryogens)."
**New text:** "These setups work reliably and don't require a supply of liquid cryogens."

**Your comment:** "Page 4, line 90: 'were performed ... - and still are poor word choice."
**Answer:** We modified the sentence and removed "still are".
**Old text:** "Indeed, much of the initial studies were performed with $lN_2$-based PHGs - and still are (Kiryutin et al., 2017; Meier et al., 2019)."
**New text:** "Indeed, IN2-based PHGs are still used in many studies (Kiryutin et al., 2017; Meier et al., 2019)"

**Your comment:** "Page 4, line 95: 'way beyond' - poor word choice."
**Answer:** We agree that it is colloquial and we removed "way" from the sentence.

**Your comment:** "Page 5, line 113:'Iron (III) oxide' - should be hydrated iron oxide or FeO(OH)."
**Answer:** We agree and changed it to the more specific name "Fe(OH)O".
**New text:** "About 1.5 ml granular Fe(OH)O (371254-50G, Sigma-Aldrich, St. Louis, U.S.A.) was filled into the coil."

**Your comment:** "Page 8, line 183: should 'norm' stand for normal pressure and

temperature (not standard)? See, for example, https://www.engineeringtoolbox.com/stp-standard-ntp-normal-air-d_772.html

**Answer:** We understand that "stand" is a better choice as an index. Therefore, we substituted "norm" by "stand" everywhere, i.e. "$p_{norm}$" by "$p_{stand}$"

**Old text:** "[...] where $T_{rt}$ is the temperature of the quantification experiment (here: 22 °C) and 'norm' stands for standard pressure and temperature values ($p_{norm} = 10^5$ pascals = 1.0 bar, $T_{norm}$ = 273.15 K) (Nic et al., 2009)."

**New text:** "[...] where $T_{rt}$ is the temperature of the quantification experiment (here: 22 °C) and 'stand' stands for standard pressure and temperature values ($p_{norm} = 10^5$ pascals = 1.0 bar, $T_{stand}$ = 273.15 K) (Nic et al., 2009)."

**Your comment:** "Page 10, line 227: '250 mm internal diameter, 360 mm outer diameter' - numbers seem too big, please check."

**Answer:** You are absolutely right! We corrected the unit "mm" to "$\mu$m".

**New text:** "The only modification was that a hollow optical fibre (Molex, part. num. 106815-0026, 250 $\mu$m internal diameter, 360 $\mu$m outer diameter) was glued to the end of the PTFE capillaries to reduce magnetic field distortions."

**Your comment:** "Page 11, line 253: 'These collisions'?"

**Answer:** We changed the paragraph in a way, that the choice of word and structure is more appropriate.

**Old text:** "The flow rate is an important parameter since it affects the collisions of $H_2$ with the catalyst in the ortho-para conversion unit (Fig. 3, A3). At optimal parameters, lN2 based PHG can provide $f_{pH_2}$ = 52 % (Fig. 2,7b). These collisions are responsible for the fast ortho-para conversion"

**New text:** "The flow rate is an important parameter since it affects the amount of $H_2$ collisions with the catalyst in the ortho-para conversion unit (Fig. 3, A3) that enables fast para-ortho-conversion. A lN$_2$-based PHG can provide $f_{pH_2}$ = 52 % at maximum

(Fig. 2,7b)."

**Your comment:** "Page 11, line 259: 'ad-hoc' isused improperly in this sentence."
**Answer:** We agree that "ad-hoc" is not the right word in this context. We modified the sentence.
**Old text:** "This flow rate was fast enough for convenient ad-hoc $pH_2$ production."
**New text:** "This flow rate was fast enough for a convenient pH2 production."

**Your comment:** "Page 12, line 269 and 273: errors and mean values should have the same number of significant figures, i.e., either 51.50(36) or 51.5(4)"
**Answer:** We totally agree and changed all results in the manuscript to the format XX,X $\pm$ X,X.
**Old text (e.g.):** "The average $f_{pH_2}$ was found to be (51.5 $\pm$ 0.36) %, corresponding to a coefficient of variation (CV) of 0.7 % (Fig. 8)."
**New text:** "The average $f_{pH_2}$ was found to be (51.5 $\pm$ 0.4) %, corresponding to a coefficient of variation (CV) of 0.7 % (Fig. 8)."

We hope, that we could address all your comments appropriately.

With kind regards,
Frowin Ellermann and Jan-Bernd Hövener

---

## Author Comment (AC3) · 11 Dec 2020

Dear James Eills,

We very appreciate your contribution to our manuscript. We implemented all of your comments. Hereby, we want to provide a point-by-point response to your comments.

**Your comment:** Title: "3D-printed" should be removed from the title or clarified as "partially 3D printed"(or similar), as this is misleading in its current form.

[Figure]

**Answer:** We agree. And to avoid this confusion modified the title as follows:
New title: "Open-source, partially 3D-printed, high-pressure (50 bar) liquid-nitrogen-cooled parahydrogen generator"

**Your comment:** Line 40: The authors might choose to also mention long-lived spin isomers in methylrotors, e.g. doi: 10.1021/ja410432f
**Answer:** Thanks for pointing out this paper! We added it to the list.
**Old text:** "Hydrogen is not the only compound that has stable or long-lived spin isomers at room temperature (rt) there are many examples: deuterium (Knopp et al., 2003), water (Mammoli et al., 2015; Meier et al., 2015), ethylene (Zhivonitko et al., 2013), and even naphthalene derivative (Stevanato et al., 2015)."
**New text:** "Hydrogen is not the only compound that has stable or long-lived spini-somers at room temperature (rt) there are many examples: deuterium (Knopp et al., 2003), water (Kravchuk et al., 2011; Vermette et al., 2019), ethylene (Zhivonitko et al., 2013) and methyl groups (Meier et al., 2013)."

**Your comment:** Line 113: Can the authors make any comment about how well the cotton wool works in preventing iron oxide flow through the system, or state how much was used for effective filtering?
**Answer:** To address this in the text we modified and added one section in the text. Now it is as follows
**New text:** "On both ends of the copper coil, cotton wool was pressed to keep the catalyst in place to protect the rest of the system from contaminations. The compressed wool insets have a length of 20 mm. Wool as a particulate filter was used before in another PHG (Du et al., 2020). During the six months of weekly use of our generator, there was no sign of a moving catalyst."

**Your comment:** Line 141: It would be helpful to briefly state how inductive and static

spark charges are avoided.

**Answer:** That's a good point! In contrast to fluids, gas does not induce charges in pipes. We modified the safety concept and added a sentence to the discussion.

**Old text:** "iv. Avoidance of inductive and static spark charges"

**New text:** "iv. Avoidance of inductive and static spark charges in the gas lines (due to conductive and groundable pipe material)"

And in discussions: "The risk for static and inductive spark charges in the gas line is low (Department of Labour of New Zealand, 1990). Nevertheless, the gas pipes can be grounded to prevent electrical charges on the parts which are in contact with $H_2$ gas."

**Your comment:** Line 265: I'm a little bit confused about the reporting of $p_{out}$, doesn't this value vary in time? Does $p_{in}$ describe the gas pressure in the catalyst region? Isn't this the only relevant pressure when considering conversion as a function of flow rate?

**Answer:** We agree on this point. $p_{out}$ is varying during filling and now we defined $p_{target}$ that is final $p_{out}$ value. We changed the whole manuscript accordingly.

**New text:**

Definition:

We refrained from including a flow meter in the setup to keep it simple and robust. Instead, we used the time $t_{p,V}$ needed to fill a cylinder of a given volume $V_0$ to a given pressure $p_{target}$...

And example:

"For the latter, the para-enrichment was found to be constant up to a flow rate of $f_r$ = 2 SLM (for $p_{in}$ = 20 bar, $p_{target}$ = 10 bar)."

**Your comment:** Lines 343-344: But the quantification is performed on a high-field

NMR system. Better to say "We have shown that routine pH2 quality control can be performed with a low-cost benchtop NMR system", or similar.

**Answer:** We agree and modified the sentence.

**Old text:** "Both values are suited for routine $pH_2$ quality control without a need for an expensive high-field NMR system."

**New text:** "These results indicate, that the routine $pH_2$ quality control can be performed with a low-cost 1 T benchtop NMR spectrometer."

**Your comment:** Lines 347-351: I don't understand the value "points loss of fpH2 per week". A relaxation time is given to describe the exponential process and this should be sufficient to understand thefpH2change each week. Unless I've misunderstood, I suggest simply removing this.

**Answer:** This value is another way to report the lifetime, but it doesn't transfer additional information. We completely removed this paragraph because it seems to be confusing, also to other referees.

**Your comment:** Line 348: "With 120 days of lifetime, Hövener et al. reported even longer values(2013)." The phrasing of this sentence is a little strange.

**Answer:** We agree and rephrased the paragraph.

**Old text:** "Feng et al. reported a lifetime in aluminium tanks of (63.7 $\pm$ 8.3) days and about 2 % points loss of $f_{pH_2}$ per week (2012). With 120 days of lifetime, Hövener et al. reported even longer values (2013).

**New text:** "The relaxation time constant in aluminium tanks was found to be (63.7 $\pm$ 8.3) days by Feng et al. (2012) and 120 days by Hövener et al., respectively (2013)."

**Your comment:** Line 349: It would be helpful to explain why vacuuming the cylinder should lead to a longer pH2lifetime (presumably due to removal of paramagnetic oxygen molecules).

**Answer:** We removed the part related to vacuuming since we did not perform a cleaning procedure with the storage bottles. However, the production protocol features a flushing routine which removes potential air contamination in the cylinder.

**Old text:** "Note, that we did not vacuum our cylinder that can increase the lifetime."

**New text:** "Note, that we did not perform any dedicated cleaning procedure for the $pH_2$ storage bottle."

**Your comment:** Figure 7: It would be informative to include an additional plot showing para-enrichment level as a function of pressure (Pin) for a 2 SLM flow rate. This is a suggestion and I do not insist on this for publication.

**Answer:** We did not observe any pressure dependency for the $p_{in}$ values we investigated (12 bar, 20 bar, 35 bar, 50 bar) and now it is mentioned in the text:

**New text:** "We also investigated $f_{pH_2}$ as a function of the $p_{in}$ pressure at a fixed flow rate: a batch was prepared for $p_{in}$ equals 12 bar, 20 bar, 35 bar and 50 bar, $p_{target}$ = 10 bar and a flow rate of 0.9 SLM. No pressure dependency could be observed. The obtained average of $f_{pH_2}$ is (52.4 ± 0.8) %."

**Your comment:** In another interactive comment, Prof. Igor Koptyug has provided two references to other parahydrogen generators described in the literature. I agree that this closely-related work should be cited.

**Answer:** The papers were published after we finished our manuscript. Nevertheless, they provide valuable information to our paper and we included both He-based PHGs into our overview table.

Thank you very much again for your extensive review, and we hope, that we could address all your comments appropriately.

With kind regards,
Frowin Ellermann and Jan-Bernd Hövener

---

## Author Comment (AC4) · 11 Dec 2020

Dear Eleonora Cavallari,

Thank you very much for your review! First of all, we implemented all your comments. Hereby, we provide a point-by-point response to the critics provided.

**Your comment:** "I find the title misleading, reading it I thought the authors provided a printable 3D-model of the parahydrogen generator, therefore I suggest to modify it."

[Figure]

**Answer:** We got the same feedback also from other referees and the new title now reads as follows:
**New title:** "Open-source, partially 3D-printed, high-pressure (50 bar) liquid-nitrogen-cooled parahydrogen generator"

**Your comment:** "The cotton wool filters used to protect the system from catalyst contamination it could be better described.
**Answer:** We agree that the information about the wool was not detailed enough. That's why we added more information now.
**Old text (e.g.):** "In both ends of the coil, the cotton wool was pressed to keep the catalyst in place and protect the rest of the system from contaminations."
**New text:** "On both ends of the copper coil, cotton wool was pressed to keep the catalyst in place to protect the rest of the system from contaminations. The compressed wool insets have a length of 20 mm. Wool as a particulate filter was used before in another PHG (Du et al., 2020). During the six months of weekly use of our generator, there was no sign of a moving catalyst."

**Your comment:** It would be interesting to know the life time if the cylinder had been vacuumed or even washed with $H_2/pH_2$ before being filled with $pH_2$, in order to verify that the value obtained using this generator is comparable with those reported previously."
**Answer:** We changed the sentence concerning the vacuuming since we did not investigate it. The life-time of $pH_2$ does not depend on PHG but purity of the storage bottle. See production protocol stage 2: Flushing storage bottle. It is the only cleaning procedure that was used.
**Old text:** "Note, that we did not vacuum our cylinder that can increase the lifetime."
**New text:** "Note, that we did not perform any dedicated cleaning procedure for the bottle. Still, the lifetime is sufficiently long to produce $pH_2$ once a week."
**Your comment:** "I don't understand the meaning of "points loss of $f_{pH_2}$. Is this value related in someway to the polarization obtainable (13) with respect to $f_{pH_2} \approx 100$ %? The authors should rephrase the sentence to clarify this concept and/or add the formula that defines it."

**Answer:** Actually, it was another way to present the lifetime. However, it was confusing other referees, too. Therefore, we decided to remove it completely since it does not contain any additional information.

Thank you again for your positive review! We hope, that we could adapt your comments as intended.

With kind regards,
Frowin Ellermann and Jan-Bernd Hövener

―――――――――――――――――――

---

## Author Response (AR1)

**Answers to comments**

*Open-source, partially 3D-printed, high-pressure (50 bar)*
*liquid-nitrogen-cooled parahydrogen generator*

**mr-2020-27**

Frowin Ellermann et al.
December 14th, 2020

**1 SC1: Igor Koptyug**

**Referee comment:** From the title, I had an impression that the generator or at least its essential parts were 3D-printed, which, as I found later, was not the case. I believe it may be a good idea to refine the paper title.
**Answer:** We understand that the title could be misleading and to avoid it the new title reads as "Open-source, partially 3D-printed, high-pressure (50 bar) liquid-nitrogen-cooled parahydrogen generator"

**Referee comment:** As always, I'm advocating the spelling of "parahydrogen" and "orthohydrogen" as single words without a dash, which I believe is the only correct way to spell them (cf. paratrooper, parabola, orthophosphoric acid; also in dictionaries, e.g.,`https://www.thefreedictionary.com/parahydrogen`).
**Answer:** We understand your point and removed the dash. Additionally, we stick to the non-italic form.
**Old text:** "para-hydrogen"
**New text:** "parahydrogen" throughout the text

**Referee comment:** The word "allotrope" in the reference to parahydrogen is acceptable in the historic context, but in fact is incorrect. By definition, allotropy is the existence of a chemical element in two or more forms, which may differ in the arrangement of atoms in crystalline solids or in the occurrence of molecules that contain different numbers of atoms (e.g., graphite, charcoal, diamond, fullerenes). Parahydrogen is this not an allotropic form of $H_2$ but rather its nuclear spinisomer.
**Answer:** We changed the word allotrope/allotropic to "nuclear spinisomer" except for the citation and the text referring to it.
**Old text (e.g.):** "The spin of hydrogen nuclei (proton) is the origin of the two allotropic forms or two spin isomers of hydrogen."
**New text:** "The spin of hydrogen nuclei (of protons) is the origin of the two nuclear spinisomer forms of dihydrogen."

**Referee comment:** line 40, naphthalene derivative (Stevanato et al., 2015) cannot be classified as spin isomer as it is not symmetric, so that the generalized Pauli principle is not applicable to it. 5) Line 39, water (Mammoli et al., 2015; Meier et al., 2015) was not enriched in the Mammoli paper experiments, or at least could not be extracted to RT. A better reference to

spin isomers of free water would be to the molecular beam separation experiments (e.g., doi: 10.1126/science.1200433 or 10.1021/acs.jpca.9b04294).

**Answer:** We rephrased the sentences and added proposed references.

**Old text (e.g.):** "Hydrogen is not the only compound that has stable or long-lived spin isomers at room temperature (rt) there are many examples: deuterium (Knopp et al., 2003), water (Mammoli et al., 2015; Meier et al., 2015), ethylene (Zhivonitko et al., 2013), and even naphthalene derivative (Stevanato et al., 2015)."

**New text (e.g.):** "Hydrogen is not the only compound that has stable or long-lived spinisomers at room temperature (rt) there are many examples: deuterium (Knopp et al., 2003), water (Kravchuk et al., 2011; Vermette et al., 2019) , ethylene (Zhivonitko et al., 2013) and methyl groups (Meier et al., 2013). Although some molecules are not symmetric and cannot be extracted at room temperature, they possess long-lived spin states of minutes (Pileio et al., 2008) and hours (Stevanato et al., 2015)."

**Referee comment:** There are a couple of very recent papers describing different parahydrogen gener-ator designs which the authors may wish to cite, doi: 10.1021/acs.analchem.0c03358doi: 10.1016/j.jmr.2020.106869

**Answer:** These papers we not yet published when we finished our manuscript. Nevertheless, we included both generators to our overview table (Table 1).

**2 RC1: Anonymous Referee**

**Referee comment:** "In particular,Page 2, line 46: '0Z axis'? Should it be Z axis?"

**Answer:** As proposed, we changed "0Z axis" to more common "Z axis" in the text

**Referee comment:** "Page 4, line 84: 'don't require much in terms of service' - poor word choice."

**Answer:** We agree and modified the sentence. Actually, the point here is that these setups don't require the supply cryogens.

**Old text:** "These setups work reliably and don't require much in terms of service (e.g. no liquid cryogens)."

**New text:** "These setups work reliably and don't require a supply of liquid

cryogens."

**Referee comment:** "Page 4, line 90: 'were performed ... - and still are poor word choice."
**Answer:** We modified the sentence and removed "still are".
**Old text:** "Indeed, much of the initial studies were performed with $lN_2$-based PHGs - and still are (Kiryutin et al., 2017; Meier et al., 2019)."
**New text:** "Indeed, lN2-based PHGs are still used in many studies (Kiryutin et al., 2017; Meier et al., 2019)"

**Referee comment:** "Page 4, line 95: 'way beyond' - poor word choice."
**Answer:** We agree that it is colloquial and we removed "way" from the sentence.

**Referee comment:** "Page 5, line 113:'Iron (III) oxide' - should be hydrated iron oxide or FeO(OH)."
**Answer:** We agree and changed it to the more specific name "Fe(OH)O".
**New text:** "About 1.5 ml granular Fe(OH)O (371254-50G, Sigma-Aldrich, St. Louis, U.S.A.) was filled into the coil."

**Referee comment:** "Page 8, line 183: should 'norm' stand for normal pressure and temperature (not standard)? See, for example,`https://www.engineeringtoolbox.com/stp-standard-ntp-normal-air-d_772.html`
**Answer:** We understand that "stand" is a better choice as an index. Therefore, we substituted "norm" by "stand" everywhere, i.e. "$p_{\mathrm{norm}}$" by "$p_{\mathrm{stand}}$"
**Old text:** "[...] where $T_{\mathrm{rt}}$ is the temperature of the quantification experiment (here: 22 ℃) and 'norm' stands for standard pressure and temperature values ($p_{\mathrm{norm}} = 10^5$ pascals = 1.0 bar, $T_{\mathrm{norm}} = 273.15$ K) (Nic et al., 2009)."
**New text:** "[...] where $T_{\mathrm{rt}}$ is the temperature of the quantification experiment (here: 22 ℃) and 'stand' stands for standard pressure and temperature values ($p_{\mathrm{norm}} = 10^5$ pascals = 1.0 bar, $T_{\mathrm{stand}} = 273.15$ K) (Nic et al., 2009)."

**Referee comment:** "Page 10,line 227: '250 mm internal diameter, 360 mm outer diameter' - numbers seem too big,please check."
**Answer:** You are absolutely right! We corrected the unit "mm" to "$\mu$m".
**New text:** "The only modification was that a hollow optical fibre (Molex, part. num. 106815-0026, 250 $\mu$m internal diameter, 360 $\mu$m outer diameter) was glued to the end of the PTFE capillaries to reduce magnetic field distortions."

**Referee comment:** "Page 11, line 253: 'These collisions'?"
**Answer:** We changed the paragraph in a way, that the choice of word and structure is more appropriate.
**Old text:** "The flow rate is an important parameter since it affects the collisions of $H_2$ with the catalyst in the ortho-para conversion unit (Fig. 3, A3). At optimal parameters, lN2 based PHG can provide $f_{pH_2} = 52$ % (Fig. 2,7b). These collisions are responsible for the fast ortho-para conversion"
**New text:** "The flow rate is an important parameter since it affects the amount of $H_2$ collisions with the catalyst in the ortho-para conversion unit (Fig. 3, A3) that enables fast para-ortho-conversion. A $lN_2$-based PHG can provide $f_{pH_2} = 52$ % at maximum (Fig. 2,7b)."

**Referee comment:** "Page 11, line 259: 'ad-hoc' isused improperly in this sentence."
**Answer:** We agree that "ad-hoc" is not the right word in this context. We modified the sentence.
**Old text:** "This flow rate was fast enough for convenient ad-hoc $pH_2$ production."
**New text:** "This flow rate was fast enough for a convenient $pH_2$ production."

**Referee comment:** "Page 12, line 269 and 273: errors and mean values should have the same number of significant figures, i.e., either 51.50(36) or 51.5(4)"
**Answer:** We totally agree and changed all results in the manuscript to the format XX,X ± X,X.
**Old text (e.g.):** "The average $f_{pH_2}$ was found to be (51.5 ± 0.36) %, corresponding to a coefficient of variation (CV) of 0.7 % (Fig. 8)."
**New text:** "The average $f_{pH_2}$ was found to be (51.5 ± 0.4) %, corresponding to a coefficient of variation (CV) of 0.7 % (Fig. 8)."

**3  RC2: James Eills**

**Referee comment:** Title: "3D-printed" should be removed from the title or clarified as "partially 3D printed"(or similar), as this is misleading in its current form.
**Answer:** We agree. And to avoid this confusion modified the title as follows:
New title: "Open-source, partially 3D-printed, high-pressure (50 bar) liquid-nitrogen-cooled parahydrogen generator"

**Referee comment:** Line 40: The authors might choose to also mention long-lived spin isomers in methylrotors, e.g. doi: 10.1021/ja410432f
**Answer:** Thanks for pointing out this paper! We added it to the list.
**Old text:** "Hydrogen is not the only compound that has stable or long-lived spin isomers at room temperature (rt) there are many examples: deuterium (Knopp et al., 2003), water (Mammoli et al., 2015; Meier et al., 2015), ethylene (Zhivonitko et al., 2013), and even naphthalene derivative (Stevanato et al., 2015)."
**New text:** "Hydrogen is not the only compound that has stable or long-lived spinisomers at room temperature (rt) there are many examples: deuterium (Knopp et al., 2003), water (Kravchuk et al., 2011; Vermette et al., 2019), ethylene (Zhivonitko et al., 2013) and methyl groups (Meier et al., 2013)."

**Referee comment:** Line 113: Can the authors make any comment about how well the cotton wool works in preventing iron oxide flow through the system, or state how much was used for effective filtering?
**Answer:** To address this in the text we modified and added one section in the text. Now it is as follows
**New text:** "On both ends of the copper coil, cotton wool was pressed to keep the catalyst in place to protect the rest of the system from contaminations. The compressed wool insets have a length of 20 mm. Wool as a particulate filter was used before in another PHG (Du et al., 2020). During the six months of weekly use of our generator, there was no sign of a moving catalyst."

**Referee comment:** Line 141: It would be helpful to briefly state how inductive and static spark charges are avoided.
**Answer:** That's a good point! In contrast to fluids, gas does not induce charges in pipes. We modified the safety concept and added a sentence to

the discussion.

**Old text:** "iv. Avoidance of inductive and static spark charges"

**New text:** "iv. Avoidance of inductive and static spark charges in the gas lines (due to conductive and groundable pipe material)"

And in discussions: "The risk for static and inductive spark charges in the gas line is low (Department of Labour of New Zealand, 1990). Nevertheless, the gas pipes can be grounded to prevent electrical charges on the parts which are in contact with $H_2$ gas."

**Referee comment:** Line 265: I'm a little bit confused about the reporting of $p_{out}$, doesn't this value vary in time? Does $p_{in}$ describe the gas pressure in the catalyst region? Isn't this the only relevant pressure when considering conversion as a function of flow rate?

**Answer:** We agree on this point. $p_{out}$ is varying during filling and now we defined $p_{target}$ that is final $p_{out}$ value. We changed the whole manuscript accordingly.

**New text:**

Definition:

We refrained from including a flow meter in the setup to keep it simple and robust. Instead, we used the time $_{tp,V}$ needed to fill a cylinder of a given volume $V_0$ to a given pressure $p_{target}$...

And example:

"For the latter, the para-enrichment was found to be constant up to a flow rate of $f_r = 2$ SLM (for $p_{in} = 20$ bar, $p_{target} = 10$ bar)."

**Referee comment:** Lines 343-344: But the quantification is performed on a high-field NMR system. Better to say "We have shown that routine pH2 quality control can be performed with a low-cost benchtop NMR system", or similar.

**Answer:** We agree and modified the sentence.

**Old text:** "Both values are suited for routine $pH_2$ quality control without a need for an expensive high-field NMR system."

**New text:** "These results indicate, that the routine $pH_2$ quality control can be performed with a low-cost 1 T benchtop NMR spectrometer."

**Referee comment:** Lines 347-351: I don't understand the value "points loss of fpH2 per week". A relaxation time is given to describe the exponential process and this should be sufficient to understand thefpH2change each week. Unless I've misunderstood, I suggest simply removing this.
**Answer:** This value is another way to report the lifetime, but it doesn't transfer additional information. We completely removed this paragraph because it seems to be confusing, also to other referees.

**Referee comment:** Line 348: "With 120 days of lifetime, Hövener et al. reported even longer values(2013)." The phrasing of this sentence is a little strange.
**Answer:** We agree and rephrased the paragraph.
**Old text:** "Feng et al. reported a lifetime in aluminium tanks of $(63.7 \pm 8.3)$ days and about 2 % points loss of $f_{pH_2}$ per week (2012). With 120 days of lifetime, Hövener et al. reported even longer values (2013).
**New text:** "The relaxation time constant in aluminium tanks was found to be $(63.7 \pm 8.3)$ days by Feng et al. (2012) and 120 days by Hövener et al., respectively (2013)."

**Referee comment:** Line 349: It would be helpful to explain why vacuuming the cylinder should lead to a longer pH2lifetime (presumably due to removal of paramagnetic oxygen molecules).
**Answer:** We removed the part related to vacuuming since we did not perform a cleaning procedure with the storage bottles. However, the production protocol features a flushing routine which removes potential air contamination in the cylinder.
**Old text:** "Note, that we did not vacuum our cylinder that can increase the lifetime."
**New text:** "Note, that we did not perform any dedicated cleaning procedure for the $pH_2$ storage bottle."

**Referee comment:** Figure 7: It would be informative to include an additional plot showing para-enrichment level as a function of pressure (Pin) for a 2 SLM flow rate. This is a suggestion and I do not insist on this for publication.
**Answer:** We did not observe any pressure dependency for the $p_{in}$ values we investigated (12 bar, 20 bar, 35 bar, 50 bar) and now it is mentioned in the text:
**New text:** "We also investigated $f_{pH_2}$ as a function of the $p_{in}$ pressure at a fixed flow rate: a batch was prepared for $p_{in}$ equals 12 bar, 20 bar, 35

bar and 50 bar, $p_{\text{target}} = 10$ bar and a flow rate of 0.9 SLM. No pressure dependency could be observed. The obtained average of $f_{\text{pH}_2}$ is $(52.4 \pm 0.8)$ %."

**Referee comment:** In another interactive comment, Prof. Igor Koptyug has provided two references to other parahydrogen generators described in the literature. I agree that this closely-related work should be cited.
**Answer:** The papers were published after we finished our manuscript. Nevertheless, they provide valuable information to our paper and we included both He-based PHGs into our overview table.

**4 RC3: Eleonora Cavallari**

**Referee comment:** "I find the title misleading, reading it I thought the authors provided a printable 3D-model of the parahydrogen generator, therefore I suggest to modify it."
**Answer:** We got the same feedback also from other referees and the new title now reads as follows:
**New title:** "Open-source, partially 3D-printed, high-pressure (50 bar) liquid-nitrogen-cooled parahydrogen generator"

**Referee comment:** "The cotton wool filters used to protect the system from catalyst contamination it could be better described.
**Answer:** We agree that the information about the wool was not detailed enough. That's why we added more information now.
**Old text (e.g.):** "In both ends of the coil, the cotton wool was pressed to keep the catalyst in place and protect the rest of the system from contaminations."
**New text:** "On both ends of the copper coil, cotton wool was pressed to keep the catalyst in place to protect the rest of the system from contaminations. The compressed wool insets have a length of 20 mm. Wool as a particulate filter was used before in another PHG (Du et al., 2020). During the six months of weekly use of our generator, there was no sign of a moving catalyst."

**Referee comment:** It would be interesting to know the life time if the cylinder had been vacuumed or even washed with $H_2/pH_2$ before being filled

with pH$_2$, in order to verify that the value obtained using this generator is comparable with those reported previously."

**Answer:** We changed the sentence concerning the vacuuming since we did not investigate it. The life-time of pH$_2$ does not depend on PHG but purity of the storage bottle. See production protocol stage 2: Flushing storage bottle. It is the only cleaning procedure that was used.

**Old text:** "Note, that we did not vacuum our cylinder that can increase the lifetime."

**New text:** "Note, that we did not perform any dedicated cleaning procedure for the bottle. Still, the lifetime is sufficiently long to produce pH$_2$ once a week."

**Referee comment:** "I don't understand the meaning of "points loss of $f_{\text{pH}_2}$. Is this value related in someway to the polarization obtainable (13) with respect to $f_{\text{pH}_2} \approx 100$ %? The authors should rephrase the sentence to clarify this concept and/or add the formula that defines it."

**Answer:** Actually, it was another way to present the lifetime. However, it was confusing other referees, too. Therefore, we decided to remove it completely since it does not contain any additional information.

[revised manuscript text omitted]